# Numerical simulations of a kilometre-thick Arctic ice shelf consistent with ice grounding observations

Edward G.W. Gasson [1], Robert M. DeConto[2], David Pollard[3] & Chris D. Clark[1]

Recently obtained geophysical data show sets of parallel erosional features on the Lomonosov Ridge in the central Arctic Basin, indicative of ice grounding in water depths up to 1280 m. These features have been interpreted as being formed by an ice shelf—either restricted to the Amerasian Basin (the "minimum model") or extending across the entire Arctic Basin. Here, we use a numerical ice sheet-shelf model to explore how such an ice shelf could form. We rule out the "minimum model" and suggest that grounding on the Lomonosov Ridge requires complete Arctic ice shelf cover; this places a minimum estimate on its volume, which would have exceeded that of the modern Greenland Ice Sheet. Buttressing provided by an Arctic ice shelf would have increased volumes of the peripheral terrestrial ice sheets. An Arctic ice shelf could have formed even in the absence of a hypothesised East Siberian Ice Sheet.

---

[1] Department of Geography, University of Sheffield, Sheffield S10 2TN, UK. [2] Department of Geoscience, University of Massachusetts, Amherst, MA 01003, USA. [3] Earth and Environmental Systems Institute, Pennsylvania State University, Pennsylvania, PA 16801, USA. Correspondence and requests for materials should be addressed to E.G.W.G. (email: egw.gasson@gmail.com)

A thick ice shelf covering the Arctic Basin during glacial stages was proposed in the 1970s, in broad analogy to the marine-based West Antarctic Ice Sheet[1–3]; building on a hypothesis originally suggested in 1888 by Sir William Thomson[4]. Despite the implications of an ice shelf for Arctic glaciology, ocean circulation, and ice volume estimates, the hypothesis received little subsequent attention due to a lack of supporting data[5–8]. The hypothesis has attracted renewed interest arising from a number of research cruises to the Arctic that have provided evidence for ice grounding in water depths exceeding 1 km, including in the central Arctic Basin on the Lomonosov Ridge[9,10]. It continues to be debated whether erosional features are indicative of grounding by isolated iceberg keels or an extensive ice shelf[9,11–14]. Recently acquired high-resolution sonar imagery have revealed sets of linear erosional features that are parallel and spatially coherent across many tens of kilometres. These features do not resemble the chaotic cross-cutting of iceberg scouring and are the strongest evidence yet for ice shelf grounding[10]. Dating of the thin sediment drape covering the erosional features indicate that grounding occurred during the penultimate glacial

maximum (140 ka, Marine Isotope Stage 6; MIS6). Although subject to age model uncertainties, the return of polynya-type conditions as evident in sea-ice reconstructions suggests that such an ice shelf would have begun to break up during the latest MIS6[15].

It is possible that large Arctic ice shelves also formed during other glacial stages, although interestingly no erosional features deeper than ~600 m water depth have yet been dated to the Last Glacial Maximum (LGM)[10,16]. Reconstructions of Arctic Ocean temperatures indicate that, following the mid-Brunhes event (~400 ka), intermediate-depth temperatures were warmer than modern during glacial stages[17,18]. One hypothesis is that a thickening halocline, driven by decreased freshwater inputs[12] and/or the formation of an Arctic ice shelf, caused the polar surface layer and inflowing warm Atlantic waters to deepen. Although there are considerable uncertainties, this could suggest that a thick ice shelf (but one that may not have grounded) formed during each of the last four glacial maxima[18]. The ice grounding locations are collated in Fig. 1a along with the inferred ice flow direction and the location of palaeo-ice streams.

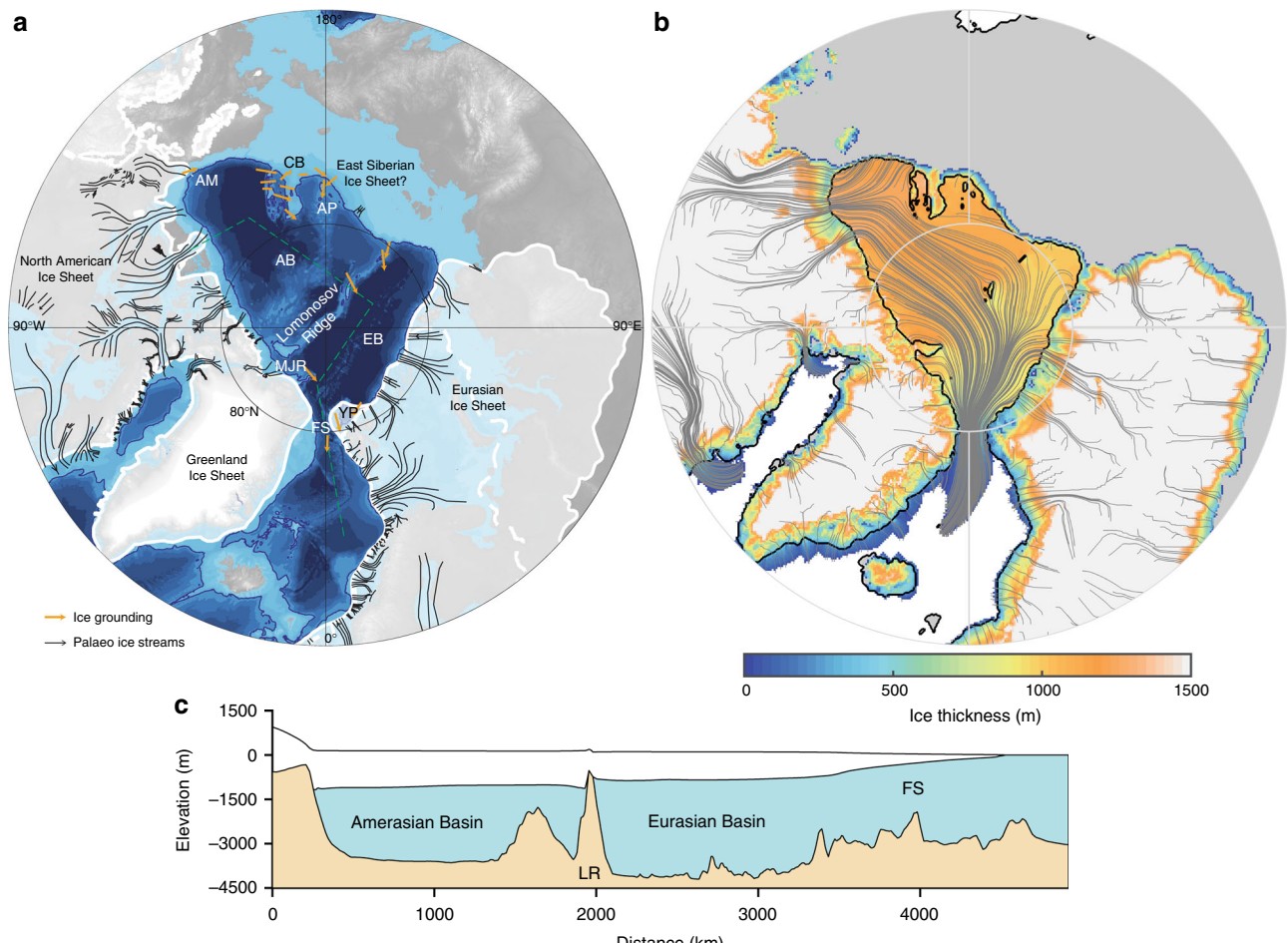

**Fig. 1** Ice shelf grounding from observations and model output. **a** Summary of location and inferred flow directions of bathymetric ice grounding features from previous publications[10,12,55–57] (orange lines and arrows). White lines shown extent of Eurasian Ice Sheet during MIS6[35] (solid line) and the LGM[30] (dashed line), and for North America at the LGM[29] (solid line). Black lines show location and flow direction of marine-terminating palaeo ice streams, for North America[58] and mapped for Eurasia using the same methodology as ref. [58], also shown is the recently discovered De Long Trough in Eastern Siberia[28]. Dashed green line is the transect shown (**c**) and dark blue line shows the 1000 m bathymetric contour. **b** Ice sheet model output showing ice thickness and streamlines, black lines show the grounding line and coastline, note areas of ice grounding on the Lomonosov Ridge, Arlis Plateau and Morris Jessup Rise. Grey shading shows land above sea level. **c** Ice sheet model output showing transect across Amerasian and Eurasian Basins and through the Fram Strait. Locations referred to in the text: AM Alaskan Margin, CB Chucki Borderland, AP Arlis Plateau, AB Amerasian Basin, LR Lomonosov Ridge, EB Eurasian Basin, MJR Morris Jessup Rise, YP Yermack Plateau, FS Fram Strait

 

Previous studies have followed statistical[19] or analytical[16] approaches to explore the plausibility of a thick Arctic ice shelf and numerical simulations have addressed the potential for an East Siberian ice shelf[20]. Here we simulate an Arctic sheet-shelf system using a hybrid shallow ice-shallow shelf model[21,22], with climate forcing provided from existing coupled climate model simulations configured for MIS6[23] but without dynamical ice-ocean coupling. A number of experiments are performed with different imposed ice sheet extents (a large versus small Laurentide Ice Sheet and with or without a hypothesised Eastern Siberian Ice Sheet[24,25]) to determine what impact the different ice source regions have on ice shelf thickness and flow direction.

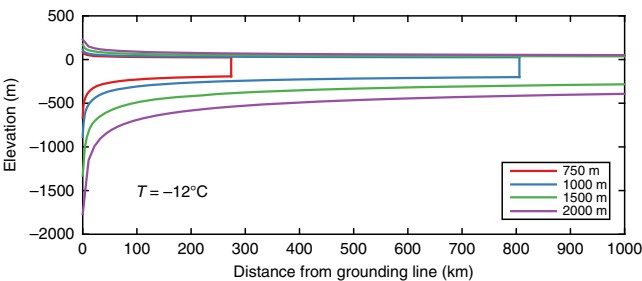

**Fig. 2** Profile of an unconstrained ice shelf. Ice shelf elevation with distance from grounding line, following analytical solution for an unconstrained ice shelf[59]. Ice flux at the grounding line is dependent on the prescribed grounding line ice thickness, as shown in the legend[60]. For a uniform ice temperature of $-12\,°C$, no ocean melting and a constant surface mass balance of 0.3 m year$^{-1}$

Although our focus is on explaining the bathymetric features on the central Lomonosov Ridge we also consider whether an ice shelf can explain other ice grounding features. A simulation is performed with the ice shelf constrained to the Amerasian Basin, in order to test the "minimum model" that the ice shelf was prevented from expanding into the Eurasian Basin due to the inflow of warm North Atlantic waters[10]. We also explore what impact the Lomonosov Ridge (i.e., through back-stress) has on the ice-shelf dynamics, by performing simulations with lowered seafloor elevation.

The ice sheet-shelf model has been used extensively for palaeo and future simulations of the Greenland and Antarctic Ice Sheets, and can successfully capture the collapse and regrowth of the marine-based West Antarctic Ice Sheet. The model is as described in ref. [21,22], unless otherwise stated. We first constrain the ice shelf calving parameterisation for simulations of the Barents-Kara Ice Sheet to its known LGM extent (see Methods and Supplementary Fig. [1]).

## Results

**Ice shelf inception.** The large area and lack of pinning points in the Arctic Basin poses a problem for the inception of an Arctic ice shelf. Although it has been suggested that bathymetric highs on the Lomonosov Ridge could act as stabilizing points[10], the ice shelf would first have to intercept the ridge, which is several hundred kilometres from the grounding lines of the North American and Eurasian Ice Sheets. Longitudinal stretching of an ice shelf leads to dynamical thinning. Analytical solutions for an unconstrained ice shelf show that an unrealistically high ice flux

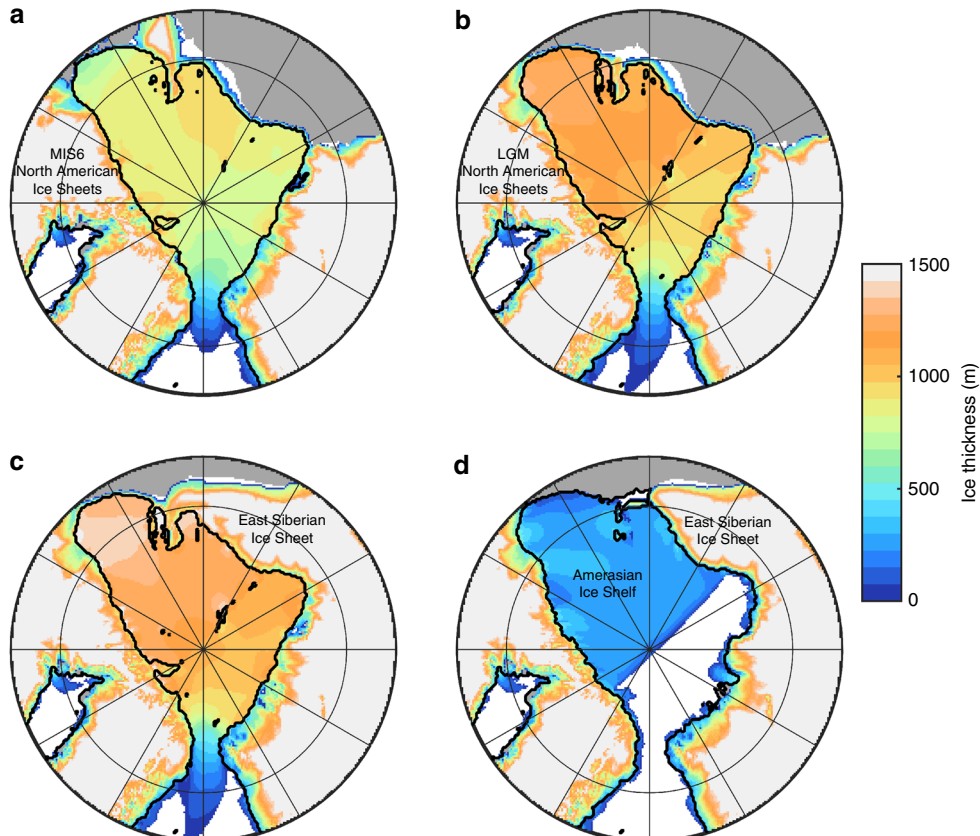

**Fig. 3** Ice shelf experiments showing simulated ice thickness. **a** Climate forcing with reduced Laurentide Ice Sheet extent[20]. **b** Climate forcing with LGM equivalent Laurentide Ice Sheet. **c** With terrestrial ice sheet able to form in Eastern Siberia. **d** Ice shelf prevented from expanding into the Eurasian Basin. Black line shows the grounding line. Model domain extends to 40 °N but is cropped in these figures. **a–d** Mean shelf thicknesses within the Arctic Basin are 833, 1070, 1173, and 254 m and floating ice volumes are 3.54, 4.42, 4.76, and 0.79 × 10$^6$ km$^3$

 

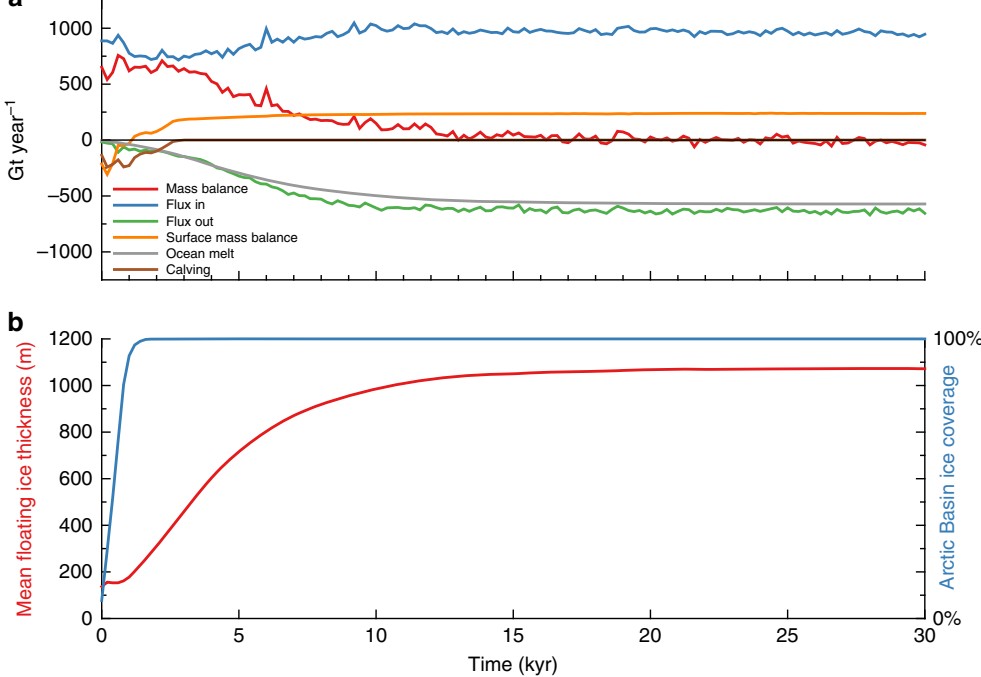

**Fig. 4** Ice shelf mass balance through time. **a** Mass balance for floating ice within the Arctic Basin, defined by location of grounding line, with a cutoff in the Fram Strait at 80˚N, for simulation shown in Fig. 3b. Calving is zero when the calving front has moved outside of the Arctic Basin. **b** Mean shelf thickness and fraction of Arctic Basin with ice cover

across the grounding line would be required to overcome dynamical thinning and maintain an ice shelf sufficiently thick to ground on the Lomonosov Ridge (Fig. 2; see also ref. [16]). However, this analytical solution does not consider lateral drag, which could thicken an ice shelf embayed in the Amerasian Basin enough to ground on the Lomonosov Ridge. We test this "minimum model" by performing an experiment with a high oceanic sub-ice melt rate in the Eurasian Basin, preventing the expansion of the ice shelf across the entire Arctic Basin. With a Eurasian Ice Sheet extending into Eastern Siberia this is the most favourable configuration of peripheral ice sheets for the growth of the ice shelf. Even with grounded ice flowing into the Arctic Basin from the Laurentide Ice Sheet and an Eastern Siberian Ice Sheet, the mean ice shelf thickness is only ~250 m (Fig. 3d). Because of dynamic thinning, this ice shelf cannot ground on the central Lomonosov Ridge and we therefore reject the minimum model.

In our simulations, a kilometre-thick ice shelf can only form with complete ice shelf cover in the Arctic Basin (Fig. 3a–c). This ice shelf forms in two stages: first there is an initial nucleation phase whereby individual ice shelves from the surrounding North American and Eurasian Ice Sheets coalesce in the central Arctic (see Supplementary Movie 1). This initial ice cover has a mean thickness of ~160 m. Once there is a complete ice-shelf cover, its thickness is balanced by flow into the basin and surface snow accumulation offsetting mass loss from basal melting and ice export through the Fram Strait. Additionally, complete ice-shelf cover reduces calving losses and the shelf is able to slowly thicken (Fig. 4a). As the ice shelf thickens, the basal melt rate increases, resulting in the ice shelf reaching an equilibrium thickness. The shelf thickens at a rate of 0.05–0.15 m year$^{-1}$, requiring 5–15 kyr to reach a sufficient thickness to ground on the Lomonosov Ridge (Fig. 4b).

The initial complete ice cover forms from the convergence of individual ice shelves from the surrounding ice sheets. In the model, this is dependent on a reduction in the calving thickness threshold, which we reduce in the model simulation in order to

correctly simulate the formation of the Barents-Kara Ice Sheet. It is uncertain whether unconstrained ice shelves from the Laurentide and Eurasian Ice Sheets could extend hundreds of kilometres into the Arctic Basin without calving. An alternative explanation is that the initial ice cover formed by another process. The basin could have become clogged with ice mélange or developed a thick sea ice cover that provided structural support[16,26,27], reducing calving and allowing an initial, thin ice shelf to form.

**Locations of ice grounding**. With the exception of the partial ice shelf (Fig. 3d), all of the ice sheet configurations tested can produce an ice shelf of sufficient thickness to ground on the central Lomonosov Ridge (Table 1). Despite the proximity of a grounded East Siberian Ice Sheet, the partial ice shelf does not ground on bathymetric highs of ~900 m depth on the Arlis Plateau, where there is evidence for past ice grounding[10,24]. This existence of an East Siberian Ice Sheet has been hypothesised in part because of the erosional features on the Arlis Plateau[20,24]. Previous numerical simulations have shown the potential for an East Siberian ice shelf flowing from grounded ice in East Siberia[20]. In our simulations, sufficient shelf thicknesses to generate grounding on the Arlis Plateau (Fig. 3a–c) can only be reached when ice is able to expand into the Eurasian Basin, forming a complete Arctic ice shelf. In these simulations, grounding occurs regardless of the presence of an East Siberian Ice Sheet (Fig. 3a, b). However, in the absence of this ice sheet the flow direction is north to south, away from the basin and opposite to that inferred from lineations and moraines on the Arlis Plateau[24]. South to north ice flow only occurs with an East Siberian Ice Sheet, which is also more consistent with a recently identified glacial trough on the East Siberian continental margin[28].

Two different extents of the North American Ice Sheets are considered in the climate model forcing (the ice sheet model margin is not constrained), one is equivalent to the LGM extent (Fig. 3b), and one with a reduced extent equivalent to the ICE5G

**Table 1 Water depth and simulated ice shelf thickness at ice grounding locations**

| ID | 1 | 2 | 3 | 4 | 5 | 6 | 7 | 8 | 9 |
|---|---|---|---|---|---|---|---|---|---|
| Location | North Alaska | Arlis Plateau | Arlis Plateau | Arlis Plateau | Chucki Borderland | Southern Lomonosov Ridge | Central Lomonosov Ridge | Southern Lomonosov Ridge | Yermack Plateau |
| Reference | Engels et al. (Area H)[57] | Niessen et al. (Fig. 4)[24] | Niessen et al. (Fig. 2a[a])[24] | Niessen et al. (Figure 2b[b])[24] | Dove et al. (Fig. 2c)[55] | Jakobsson et al. (Fig. 2a, b)[10] | Jakobsson et al. (Fig. 2c)[10] | Jakobsson et al. (Area B)[12] | Jakobsson et al. (Area E)[12] |
| Latitude | 70.8°N | 75.5°N | 76.5°N | 76.3°N | 76.9°N | 81.5°N | 85.0°N | 86.7°N | 80.2°N |
| Longitude | 144.2°W | 178.7°W | 178.5°W | 180.0°W | 164.8°W | 142.0°E | 152.0°E | 54.0°W | 6.8°E |
| Modern bathymetry (m) | 458 | 902 | 917 | 1107 | 575 | 879 | 807 | 810 | 621 |
| Bathymetry - ice shelf depth (m): | | | | | | | | | |
| Fig. 3a | 0 | 22 | 0 | 147 | 0 | 0 | 0 | 0 | 136 |
| Figs. 3b, 1b | 0 | 0 | 0 | 36 | 0 | 0 | 0 | 0 | 105 |
| Fig. 3c | 0 | 0 | 0 | 0 | 0 | 0 | 0 | 0 | 89 |
| Fig. 3d | 299 | 611 | 552 | 757 | 106 | 517 | 428 | n/a | n/a |

Exact locations are shown in Supplementary Fig. 4. Modern bathymetry is from ETOPO1 dataset, calculated at the resolution of the ice sheet model (20 × 20 km); therefore some features may have been smoothed. Modern-day water depth is shown, note that for all simulations sea level is lowered by 120 m. Distance between the ice shelf base and ocean floor for simulations shown in Fig. 3; when equal to zero the ice shelf has grounded
[a]Shallower glacial lineations on Arlis Plateau
[b]Deeper glacial lineations on Arlis Plateau

reconstruction at 13 ka (Fig. 3a; following ref. [23]). For the LGM, the extent of the major ice sheets is well known, constrained by a large volume of dated geomorphological and geological evidence[29,30]. For prior glaciations such as MIS6, the evidence-base in North America is sparse and its geographic spread is insufficient to develop a continent-wide reconstruction of ice extent at this time. But it is known for example that during MIS6, that the southern Laurentide Ice Sheet extended some 200 km further south in the US state of Illinois than during the LGM[31]. In Ohio the MIS6 extent is a short distance (10–20 km) further south[32]. In Minnesota, pre-LGM glacial deposits also extend beyond the known LGM ice limits[33]. However, this picture of greater MIS6 extent is not repeated around the rest of the perimeter of the ice sheet, likely because the ice sheet was smaller than during the LGM and the evidence has been erased or obscured. The lack of icebergs being discharged through the Hudson Strait during MIS6 would indicate that the North Atlantic sector of the Laurentide Ice Sheet had a reduced extent (or different dynamics) relative to the LGM. Finally, the LGM sea level lowstand was likely lower than during MIS6, with one estimate proposing a difference between the two glacial stages of $21 \pm 14$ m[34]. Because the MIS6 Eurasian Ice Sheet had a greater extent than during the LGM[35], it is likely that the North American Ice Sheets had a smaller volume during MIS6[23,36].

The simulation with reduced North American Ice Sheets has two major influences on the formation of the Arctic ice shelf. Firstly, there is reduced ice flow into the basin (460 Gt year$^{-1}$ total input into the Arctic Basin, compared with 950 Gt year$^{-1}$ for the larger North American Ice Sheets) because the western most ice streams, the MacKenzie Trough and Amundsen Gulf Ice Streams, are less active. However, this reduced flow into the basin is partially offset by higher surface accumulation on the Arctic ice shelf (370 Gt year$^{-1}$ compared to 240 Gt year$^{-1}$). The lower elevation and extent of the North American Ice Sheets generates a shift in atmospheric planetary waves, increasing temperatures and precipitation over the Eurasian Ice Sheet and leading to cooler temperatures over Eastern Siberia. The smaller North American Ice Sheets also generate increased precipitation in the Arctic, in particular in the Amerasian Basin[23]. Although these two effects partially cancel out, the reduction in ice flow from North American ice streams results in a mean shelf thickness of 830 m, compared with 1070 m for the thicker LGM-equivalent North American Ice Sheets. Despite the reduced shelf thickness, this ice

shelf still grounds at most of the locations where there is evidence for grounding (Fig. 3a; Table 1).

No evidence for deep ice grounding has yet been found for the LGM, although it is possible that a thinner, ungrounded ice shelf also existed during the LGM[10,18] or that ice grounding features have yet to be discovered. The shorter duration of the LGM compared with MIS6 is one explanation as to why evidence for ice grounding at the LGM is lacking[10]. Another possible factor is the different configurations of the peripheral grounded ice sheets, with a less easterly expansion of the Eurasian Ice Sheet, which may have delivered less ice to the Arctic Basin and prevented a thick ice shelf from forming[12]. To test this we have performed experiments with an LGM climate forcing[37] and with the Eurasian Ice Sheet restricted to its reconstructed LGM extent[30] (Supplementary Fig. 3).

In these LGM simulations, a thick ice shelf is still able to form in the Arctic Basin, which is not supported by the current geological evidence. Although the reduced volume of the Eurasian Ice Sheet results in greatly reduced ice flow into the Arctic Basin from that sector, this is compensated for by increased flow from the Laurentide Ice Sheet; likely caused by lower longitudinal stress from the reduced Eurasian Ice Sheet flow. The surface mass balance in the Arctic Basin is lower during the LGM (110 Gt year$^{-1}$), relative to the MIS6 simulations (240–370 Gt year$^{-1}$). Overall, this results in a lower equilibrium thickness (940 m) of the ice shelf, relative to a simulation with an MIS6 equivalent Eurasian Ice Sheet (1070 m). However, our simulated LGM ice shelf would have left ice grounding traces. These simulations have identical ocean forcing and the same duration as the MIS6 simulations. Given the strong sensitivity of the ice shelf thickness to changes in the ocean melt rate (Table 2), we suggest that differences in ocean circulation may explain why a thick ice shelf formed during MIS6 and not the LGM. The different configurations of the terrestrial ice sheets may have played a secondary role.

**Ice shelf dynamics**. Once the ice shelf pins on the Lomonosov Ridge, the resulting buttressing influences the ice shelf dynamics. An ice rise forms in the central Arctic where the shelf intercepts the Lomonosov Ridge (Fig. 5a). We find two principle flow regimes, a slightly thicker, slower flowing ice shelf in the Amerasian Basin and a thinner, faster flowing shelf in the Eurasian Basin, with increasing velocities towards the Fram Strait calving region; this is similar to that inferred from analytical

**Table 2 Summary of experiments and Arctic ice shelf mass balance**

| Figure | Climate | Eurasian Ice Sheet | Ocean factor | $h_{calv}$ (m) | Mean shelf thickness (m) | Shelf volume ($10^6$ km$^3$) | Flux in (Gt year$^{-1}$) | Flux out (Gt year$^{-1}$) | SMB (Gt year$^{-1}$) | Ocean melt (Gt year$^{-1}$) |
|---|---|---|---|---|---|---|---|---|---|---|
| 3a | MIS6 (T2) | Svendsen04 | 1.0 | 100 | 833 | 3.54 | 460 | −470 | 370 | −380 |
| 3b, 1b, 5a | MIS6 (T1) | Svendsen04 | 1.0 | 100 | 1070 | 4.42 | 950 | −660 | 240 | −570 |
| 3c | MIS6 (T1) | Unconstrained | 1.0 | 100 | 1173 | 4.76 | 1110 | −710 | 230 | −660 |
| 3d | MIS6 (T1) | Unconstrained | 1.0 | 100 | 254 | 0.79 | – | – | – | – |
| – | MIS6 (T2) | Unconstrained | 1.0 | 100 | 917 | 3.88 | 660 | −600 | 370 | −450 |
| 5b | MIS6 (T1) | Svendsen04 | 1.0 | 100 | 1029 | 4.32 | 1020 | −720 | 240 | −540 |
| S2a | MIS6 (T2) | Svendsen04 | 0.5 | 100 | 1052 | 4.30 | 490 | −640 | 370 | −270 |
| S2b | MIS6 (T2) | Svendsen04 | 2.0 | 100 | 617 | 2.68 | 410 | −320 | 360 | −470 |
| S2c | MIS6 (T2) | Svendsen04 | 4.0 | 100 | 433 | 1.89 | 440 | −190 | 350 | −540 |
| S2d | MIS6 (T2) | Svendsen04 | 8.0 | 100 | 253 | 1.09 | 400 | −80 | 260 | −500 |
| S1a | LGM | Hughes16 | 1.0 | 150 | – | – | – | – | – | – |
| S1b | LGM | Hughes16 | 1.0 | 125 | – | – | – | – | – | – |
| S1c, S1d | LGM | Hughes16 | 1.0 | 100 | – | – | – | – | – | – |
| S3 | LGM | Hughes16 | 1.0 | 100 | 940 | 3.93 | 940 | −640 | 110 | −460 |

All ice sheet parameters are as ref. [22], unless otherwise shown. Small differences in surface mass balance (SMB) between some simulations that have the same input climate are caused by different areal extents of the ice shelf. Note that variability in the flux of ice into and out of the Arctic Basin due to ice stream activity (see Fig. 4) means that mass balance values may not sum to zero, although the ice shelf thickness has reached equilibrium. For ease of interpretation, mass balance values are rounded to the nearest 10 Gt year$^{-1}$. Svendsen04 is the >140 ka Eurasian Ice Sheet extent of ref. [61] and Hughes16 is the 'best-estimate' maximum extent of the Eurasian Ice Sheet during the last glacial cycle, from ref. [30]
T1 is a climate simulation with LGM equivalent North American Ice Sheets, T2 is a climate simulation with approximate MIS6 North American Ice Sheets, following Ref. [23]

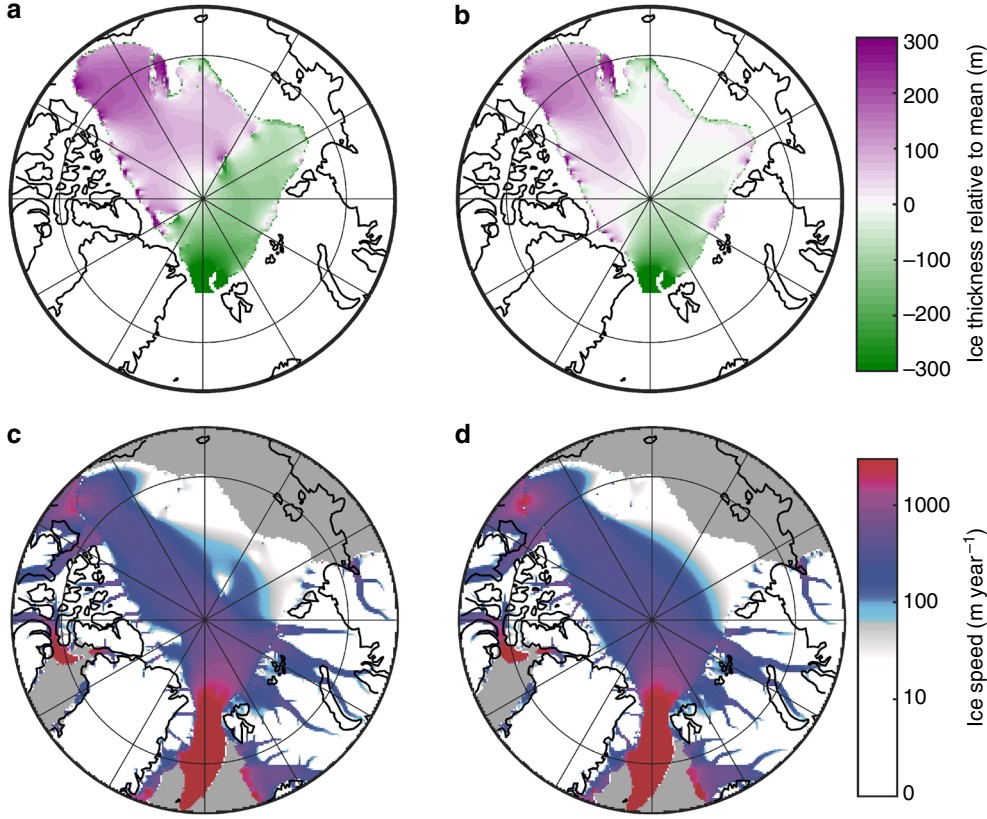

**Fig. 5** Impact of Lomonosov Ridge buttressing on ice shelf dynamics. **a, c** are the same as simulations shown in Fig. 3b, **b, d** is an identical simulation with the Lomonosov Ridge removed, location of Lomonosov Ridge is shown in Fig. 1a. The mean shelf thicknesses are comparable between the simulations (1070 and 1029 m for **a** and **b**, respectively)

approaches[16]. We perform an additional simulation with the Lomonosov Ridge removed (Fig. 5b, d). We find that these flow regimes develop even in the absence of the Lomonosov Ridge and are caused by the Arctic geometry and the location of the Fram Strait. With the shelf pinned on the Lomonosov Ridge, these flow regimes become more pronounced with an increased thickness gradient between the two basins. As highlighted by the simulations without a Lomonosov Ridge, in which a thick ice shelf is still

able to form, the bathymetric highs in the central Arctic may be less critical to the formation of an Arctic ice shelf than previously suggested[10].

The presence of an Arctic ice shelf influences the ice streams feeding into the basin and the peripheral terrestrial ice sheets. Buttressing provided by the ice shelf results in the advance of the grounding line and thickening of the surrounding ice sheets (Fig. 6). Comparing simulations with and without an Arctic ice

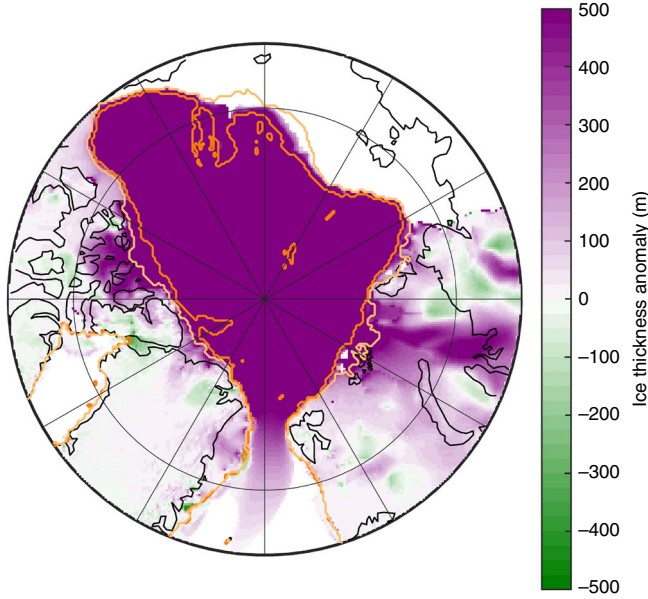

**Fig. 6** Impact of an Arctic ice shelf on grounding line position and grounded ice thickness. As simulation shown in Fig. 3b, showing thickness difference relative to a simulation without an Arctic ice shelf. Light orange line shows grounding line position in absence of an Arctic ice shelf, dark orange line is grounding line with an Arctic ice shelf. Note thickening of the terrestrial ice sheets, in particular in ice stream locations

shelf, this causes an increase in volume of the grounded ice sheets of $1.4 \times 10^6$ km$^3$, a sea level equivalent change of 3.5 m. The grounding line extends towards the Arctic Basin, with this extension especially pronounced for the Laurentide Ice Sheet.

## Discussion

The volume of the simulated Arctic ice shelf varies between 3.5 and $4.8 \times 10^6$ km$^3$, equivalent to 120–170 % of the modern Greenland Ice Sheet. Through buttressing it also leads to an increase in the volume of the peripheral grounded ice sheets (Fig. 6). With the exception of its impact through buttressing, the ice shelf would have minimal direct impact on sea level. However, many sea level reconstructions for MIS6 are proxy-based estimates calculated from the oxygen isotope composition of seawater ($\delta^{18}O_{sw}$), which will increase with the presence of such a large floating meteoric ice mass[2]. There is a 1.66‰ offset in $\delta^{18}O$ of benthic foraminifera between the Holocene and MIS6[26], which includes contributions from decreasing deep sea temperature and changes in $\delta^{18}O_{sw}$. Assuming a simplified constant ice shelf $\delta^{18}O$ composition of −40‰, the Arctic ice shelf would increase $\delta^{18}O_{sw}$ by 0.11–0.14‰, reducing the magnitude of MIS6 sea level lowstand estimates by up to 14 m.

The presence of an Arctic ice shelf may help to explain existing discrepancies between sea level reconstructions for MIS6[34,36]. As mentioned previously, there is a 21 ± 14 m offset in sea level between MIS6 and the greater sea level fall of the LGM. However, this sea level offset is smaller in deep-sea $\delta^{18}O_{sw}$ records[38], compared with sea level reconstructions based on water residence times in the semi-enclosed Mediterranean Sea and Red Sea[34,36,39]. The formation of an Arctic ice shelf would influence the Mediterranean and Red Sea reconstructions, which are also affected by changes in $\delta^{18}O_{sw}$. However, due to the larger amplitude of the glacial to interglacial $\delta^{18}O$ shift for the semi-enclosed basin records, this bias would be larger in the deep-sea $\delta^{18}O_{sw}$ reconstructions. The sea level offset, and the discrepancies between the sea level records, can be explained by the formation of a large ice

shelf during MIS6, but would require negligible floating ice mass during the LGM[34].

The significance of an Arctic ice shelf to ocean circulation and northern hemisphere glacial climates has been noted previously, both through the direct impact of a permanent thick ice cover on the atmosphere and ocean below, and additional impacts on ocean circulation during ice shelf break up[6,24]. The routing of meltwater from terrestrial ice sheets into the Arctic and the subsequent release of large amounts of freshwater through the Fram Strait has the potential to disrupt the thermohaline circulation in the North Atlantic[40]; an Arctic ice shelf is an additional source of freshwater. More work is needed to determine whether the magnitude and rate of ice shelf break-up would be sufficient to influence ocean circulation. It is noteworthy that because of the thickness and spreading rate of the ice shelf, once the calving front has retreated back into the Arctic Basin, calving laws suggest that break-up of the shelf would have been rapid[41].

## Methods

**Ice sheet/shelf model**. The ice sheet model used is a hybrid model that combines the scaled shallow ice and shallow shelf equations[21], we briefly describe the model, which is setup as documented in ref.[22] unless otherwise mentioned below. Vertical shearing and basal stress balances the gravitational driving stress for slow flowing, grounded ice regimes, whereas horizontal stretching dominates in fast flowing grounded ice regimes and floating ice shelves. The combined shallow ice-shallow shelf (SIA-SSA) equations account for both flow regimes, with computational demand reduced in slow-flowing areas with limited basal sliding that can be satisfactorily modeled with solely SIA dynamics. The grounding line can freely migrate using a sub-grid parameterisation that relates ice velocity to ice thickness at the grounding line[42,43]. This parameterisation eliminates the resolution-dependence of ice flow at the grounding line (allowing relatively coarse grids to be used for long-duration simulations) and overrides the ice velocities computed by the SIA-SSA at the grounding line. The parameterisation also accounts for buttressing by downstream islands, pinning points or side shear that can generate back stress, reducing grounding line velocities. Basal sliding coefficients ($C$) vary based on the distribution of deformable sediments, with a sediment map[44] used to specify regions of hard bedrock ($C = 10^{-10}$ m a$^{-1}$ Pa$^{-2}$) and deformable sediment ($C = 10^{-6}$ m a$^{-1}$ Pa$^{-2}$).

Ice-shelf calving is a combined parameterisation dependent on flow divergence, velocity, and hydrofracture in climates producing surface meltwater or liquid precipitation[45]. An added calving constraint is imposed which restricts the minimum thickness of ice shelves ($h_{calv}$). In Antarctic model runs this parameter is needed to prevent the seaward extension of the Filchner-Ronne and Ross ice shelves beyond their modern-day calving fronts, however the parameter prohibits the regrowth of the West-Antarctic Ice Sheet from a deglaciated state. Experiments conducted here show that the default calving thickness threshold (150 m) also prevents the growth of the Barents-Kara Ice Sheet (BKIS) and formation of an ice sheet in Hudson Bay during the LGM (see Supplementary Fig. 1); this result is not dependent on the ocean melt rate. Regrowth of the West-Antarctic Ice Sheet either requires removal of the calving thickness constraint[45] or the reduction of calving in confined embayments calculated from the arc angle to open ocean[22,43]. This is attributed to clogging with ice mélange and/or sea ice[22]. For the Arctic simulations the arc angle reduction in calving is not used, as it would treat the entire Arctic Basin as a confined embayment. We therefore lower $h_{calv}$ until we are able to form a BKIS. A reduction in $h_{calv}$ to 100 m allows formation of the BKIS at the LGM and this value is used for all subsequent experiments. Additional simulations have shown that with $h_{calv}$ equal to 150 m, an Arctic ice shelf is unable to form. Simulation of ice mélange and its impact on calving is the subject of future research. A new mechanism for the structural failure of large ice cliffs is included in the ice sheet model[45], although this has minimal impact on the results for these glacial maximum simulations.

The ice margin of the Eurasian Ice Sheet is constrained in a number of the experiments presented; due to the lack of a reliable reconstruction of the MIS6 North American Ice Sheets, they are always unconstrained in the ice-sheet model. The Eurasian margin is restricted to either the 140 ka ice margin[35] or the LGM[30] for sensitivity experiments. For grid cells within ±40 km of the margin a soft constraint is imposed and accumulation is set to zero, further outside of the margin a hard constraint is imposed and a strong ablation is also added[46]. The impact of this constraint is clearly visible in simulations without ice-margin limits. Without this constraint the Eurasian Ice Sheet will extend into Eastern Siberia with LGM climate forcing. However, this result may be climate model dependent, with expansion not found in similar simulations performed using the IPSL CM5a climate model[47].

The model domain extends to 40°N, although the model output is cropped in some of the figures presented here. For efficiency, we first spin-up the model from ice-free conditions for 45 kyr at a resolution of 40 × 40 km, before increasing the

resolution to 20 × 20 km for another 5 kyr. During this spin-up phase, ice shelves are prevented from forming, with only the terrestrial ice sheets developing. All subsequent simulations are restarted from this full-glacial state and run for 30 kyr at a resolution of 20 × 20 km.

**Climate forcing.** Surface temperatures and precipitation from the Community Climate System Model, version 4 (CCSM4) are used to calculate the surface mass balance. These simulations do not include an Arctic ice shelf and there is therefore no feedback from the formation of an ice shelf on the ocean and climate system. A thick ice shelf cover would reduce heat exchange between the ocean and atmosphere, relative to a thin sea ice cover, and also lower surface air temperatures due to the elevation lapse rate effect[48]. These existing simulations are with either LGM[37] or MIS6[23,49] boundary conditions (astronomical configurations, ice sheet elevation, and greenhouse gas concentrations). Two separate MIS6 simulations are used with different configurations for the North American Ice Sheets, either with the same extent as LGM, or with a reduced extent and elevation required to maintain sea level budgets given the larger extent of the Eurasian Ice Sheet during MIS6[23,35]. The reduced North American Ice Sheets are based on the ICE5G model for 13 ka[50], although the actual extent of the North American Ice Sheets during MIS6 is uncertain. The differences in the extent and elevation of the North American Ice Sheets has a significant impact on the climate of the Arctic and Eurasia[23]. Surface temperatures are lapse rate corrected to account for differences in GCM elevation and the ice-model surface elevation. The surface temperatures from CCSM4 are also anomaly (bias) corrected based on errors between a modern-day control simulation and an observational temperature dataset. Precipitation is not anomaly corrected. Bias correcting leads to a greater agreement between the simulated LGM ice thicknesses and the ICE6G model. Surface ablation is calculated using the positive-degree day method. Because we are using a constant climate forcing, there may be more rapid transitions that are not currently accounted for.

The MIS6 climate forcing produces a strong negative surface mass balance along the North Alaskan margin. This can result in the export of shelf ice via re-grounding, followed by subsequent surface melting and instances with spurious ice flow. As there is no evidence for export of ice/meltwater via the North Alaskan margin, we limit the surface ablation for grid cells with grounded ice to no greater than 0.5 m year$^{-1}$ in that sector (125–180°W, 68–90°N). A similar adjustment is required to allow glacial inception on Svalbard for ice sheet model runs starting from ice-free conditions, with an initial ablation limit of <0.2 m year$^{-1}$ required in that sector (8–32°E, 76–82°N), this does not affect floating ice cells.

**Ocean forcing.** Sub ice-shelf basal melt rates are clearly of great importance to the existence and equilibrium thickness of Arctic ice shelves, however they are poorly constrained today and worse in the past. Proxy reconstructions show that the Arctic Ocean structure differed during glacial intervals, with a deepening of warm Atlantic waters[17]. The presence of an ice shelf in the Arctic could affect the ocean structure and circulation[10], however no simulations have yet examined these effects on basal melt rates. We therefore adopt a conservative approach to sub-ice melting, calculating basal melt rates based on a last glacial simulation[51] that has been calibrated for Antarctica[52] and exploring model sensitivity through large adjustments (50–800%) of a melt enhancement factor ($O_{fac}$). Changes in the basal melt rate can lead to variation in the ice shelf equilibrium thickness of 100 s of metres (see Table 2 and Supplementary Fig. 2). For the default melt enhancement factor, used in all experiments in the main manuscript, basal melt rates at 1000 m depth are ~0.16 m year$^{-1}$ in the central Arctic. This is comparable to the basal melt rates of <0.2 m year$^{-1}$ estimated by ref. [10] using a conceptual oceanographic model. Ocean melt rates are parameterized using a quadratic dependence on the difference between simulated ocean temperatures and the freezing point of seawater[21,53,54]. Because the freezing point of seawater increases with ocean depth, this leads to an increase in the melt rate with depth, and hence an increase in the melt rate as the ice shelf thickens. In experiments where the ice shelf is prevented from expanding into the Eurasian Basin we impose a large increase in the melt enhancement factor for the Eurasian Basin.

**Data availability.** The data and model output that support the findings of this study are available from E.G. upon reasonable request.

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

## Acknowledgements

We thank Florence Colleoni for providing climate model data. We also thank Jeremy Ely for providing the unpublished mapped Eurasian palaeo ice streams shown in Fig. 1a. E.G. is supported by a University of Sheffield fellowship. We thank the three reviewers for their valuable comments and suggestions.

## Author contributions

E.G. conceived the project, performed simulations and wrote the paper. R.D., D.P. and C.C. contributed to the discussion of the results and commented on the manuscript.

## Additional information

**Competing interests:** The authors declare no competing interests.

