## [Peer Review File(PDF 173 kb) · Nature Communications]

Reviewers' comments:

Reviewer #1 (Remarks to the Author):

Review of Gasson et al. NatureComm

"Numerical simulations of a kilometre-thick Arctic ice shelf consistent with ice grounding observations"

General

This is a really good new modeling study of the hypothesized Arctic Ocean-wide MIS thick ice shelf, and also perhaps a smaller ice shelf during the LGM. It is well written and the model results are plausible. But in places in the paper, the authors raise tantalizing topics (see specifics listed below) like rates of sea level rise, ocean circulation impacts of ice-shelf melting (ie page 8), calibration of foram O18 to ice volume, E Siberian Sea ice, etc, but they never really delve into them. These topics are huge in the eyes of NatComm readership. I would like to see more discussion of the implications of complete ice shelf cover during MIS 6, and post MIS 6 disintegration and likewise for other glacials, other than implications for the ice shelf modeling itself developed for Antarctica. The authors also do not address the question of WHEN these massive ice sheet/shelf systems began: end Pliocene? Mid Pleistocene Transition, Mid-Brunhes Event? [a new paper on the MBE is coming out any day relevant to the MBE threshold in the Arctic]. Finally are there any implications for modeling past and present Antarctic ice shelves and margins, since, if their Antarctic model is supported by all the empirical field evidence in the Arctic, this is an important conclusion. In Figures, Label ice sheets in Fig 1b, spell out Lomonosov Ridge, Amerasian, Eurasian Basin in Fig 1c. Fig 2 & fig. 4 have NO labeling, please add a little.

In sum, perhaps the authors are understating the implications of their results for rapid deglaciation, SL rise, ice shelf collapse etc. Is the time framework shown in Figure 3 accurate, or, could there be more abrupt transitions that simply are not accounted for in the model structure. Recommendation: Publish with minor revision.

Specifics

Line 34. Dating the break up at the end of MIS 6 is a hard thing to do [ie date], although Stein et al gave their best shot. This is critical and LGM ice break up is much better known due to the many C14 dated studies showing a widespread glacial unconformity and C14 dates mainly starting ~ 15 ka, the B/A most likely (Polyak et al. 2007, 2009, Poirier et al. 2012, Gemery et al 2017 Climate of the Past and many studies cited in these.

Line 38 are the ice streams in Figure 1 from Stokes' studies? Please cite here.

Line 42 spelling Arctic

Line 44. Though it may be covered later, it would be useful to say here up front if these models used include sub-ice ocean forcing or not.

Line 47. More recent sea level and GIA evidence for E Siberian ice is found in Cronin et al. 2017 Climate of the Past (See also line 139)

Line 69 whose N American and Eurasian grounding lines are you using here?

Line 89-90 say ice shelf thickness or "its thickness" so as not to confuse with continental shelf

Line 96-99. Above there were citations to others favoring an LGM ice shelf cover, but here the idea is dismissed fairly quickly.

Line 103 you mean "which we reduced in the model simulation... ??

Line 154 we need more precise data on LGM & MIS 6 minimum sea level, from submarine regions.

Line 172 the volume ... varies... ??

Line 178 awkward sentence

Line 180-181, is this corrected for deep bottom temperature, a large source of MIS 6 oxygen isotope variability/signal.

Line 198- PLEASE Say how rapid and do estimates jive with rates of SL rise?

Reviewer #2 (Remarks to the Author):

Gasson et al. investigate the possibility of a thick Arctic ice shelf covering the entire Arctic basin during the penultimate glacial maximum (and maybe also during the LGM). They do this by using an SSA-SIA model with a number of different boundary conditions/forcings. Some of these stem from climate models (e.g. the surface mass balance) and paleo-reconstructions (e.g. the ice-sheet extents), others are treated in a more ad-hoc way (e.g. ocean melting and calving thresholds). Within this [huge] parameter space, they find stable configurations for such a large ice shelf. A key constraint for evaluating the different model runs are recently discovered geomorphological features at the Lamasov ridge (and on other ridges equally far away from the continental grounding line) which are supposedly caused by erosion from a locally re-grounded ice shelf. Smaller ice-shelves (i.e. restricted in size to the Amerasian basin) would not develop the required thickness for re-grounding, which is used as an argument to rule out a previously hypothesized 'minimum-model'.

I have read this paper with great interest, and without doubt the question whether or not such a large ice shelf has existed has important consequences for a number of disciplines such as glaciology, geomorphology, oceanography and sea-level reconstructions. The timely manner of this problem is demonstrated by a number of publications (e.g. Stein et al. 2017, Jakobsen et al. 2016, Nielsse et al. 2017) which have all been published recently. The merit of this study over previous modelling approaches (e.g. Nielsse et al. 2017, TC) is that it uses a real geometry and observationally constrained forcings which surely is a complicated task and I compliment the authors for achieving this.

I encourage the authors to put more effort into highlighting the critical mechanisms and to order them in importance (if possible). At the moment, the paper partially invokes the impression that some scenarios have been tailored (e.g. by reducing the calving threshold) so that such a large ice shelf can exist and explain the observed glacial landforms. While this in itself is also an important result, it undersells the potential of the study for other researchers.

The different model scenarios must be explained more clearly (i.e. which parameters have been varied simultaneously or individually and over what range). Currently I fear that the study will not be reproducible with the information supplied in the text. Nature Communications has no tight space limits so more plots/text can easily be added.

Below, I have a number of comments/questions (some of which may reflect my inexperience with paleo ice sheet modelling). These comments should be considered and I hope that you will find them helpful in improving the current version of the paper.

Kind regards,
Reinhard Drews, University of Tübingen, Germany

General Comments

1. The reduction of the calving threshold seems a necessity to grow this large ice shelf. It is motivated in a few lines (l. 60-62) with the BKIS which does – in this model -- not grow to its known LGM extent. However, does the reduction of the calving threshold have any physical motivation (other than maybe accounting for sea-ice cover/ice mélange). Why would calving be different in the Arctic than in the Antarctic? How can

you tell that the too small BKIS extent is not due to other factors such erroneous forcing (e.g. a too small surface mass balance or too high ocean melting?). I suggest to argue more strongly that the reduction of the calving threshold is physical reasonable and is not used as a tuning parameter correcting for a bias in the input data (or for the simplified model physics of the SIA-SSA model).

2. A sensitivity study has been performed, but it appears scattered throughout the text and important information is missing (or I missed it). Parameters include bathymetry with and without LR, melt rates with enhancement factor ranging from 0.5-8, climate input from LGM and PGM, two different extends of the Laurentide Ice Sheet, and existence/absence of an East Siberian Ice Sheet. However, it is unclear to me which parameters were varied in which run. I suggest to summarize the sensitivity studies in a table, and to name the different model runs correspondingly, otherwise it is very hard to get an overview of what was done. How many model runs were done in total?
3. The previous points highlights are more important issue: Which of all these parameters is most critical? Which one is the most ill-constrained? (Is it the missing calving-law, the ice-ocean coupling, the bathymetry?). This paper currently gives little outlook/motivation for other researchers to redirect their research focus.
4. The authors loosely use the term “pinning” for the grounding of the ice shelf on LR and I think that they mean by that the development of ice rumpled (as opposed to ice rises). However, what basal friction coefficient has been assumed for the LR and does this value matter? As a function of this, what are the basal sliding velocities? Pinning points in Antarctica can be near stagnant (e.g. Berger et al., 2016, The Cryosphere) or almost fully overridden (e.g. the Doake Ice Rumples). At which end of the spectrum do you see the LR? Without those details the statement that LR has “limited impact on ice-shelf dynamics (l. 14)” is hard to assess.
5. All but the (ruled-out) minimum-model configurations lead to grounding on LR ridge (l. 133), yet there is no evidence that grounding occurred during the LGM (l. 96). This conflict needs to be resolved. Does it mean that your bracket for ocean melting is too low on the upper end?
6. How important is the surface mass balance to sustain the ice shelf (e.g. compared to the basal melt rates)? How reliable are those values, and how does the existence of the large Arctic ice shelf impact the surface mass balance rates (is there some kind of Feedback mechanism?)
7. Do the ocean melt rates applied here bracket the estimations done by Jakobsson et al. (NComms, 2016)? Melt rates of max 1.27 m a^{-1} at 1000 m depth seem not overly high (although an ocean enhancement factor of 8 sounds high). It is important to put this into perspective with other studies because ocean melting is clearly critical to ice-shelf stability.
8. I suggest to show all the input parameters of a minimum (smallest ice sheets, highest melting, highest calving threshold, no East Siberian Ice Sheet) versus a maximum (largest ice sheets, smallest basal melting, smallest calving threshold, East Siberian Ice

Sheet) scenario in a supplementary Figure. This would also make Fig. 1 more accessible (which contains some of this information but not all).

l. 16: "Significance" for the hypothesize East Siberian Ice Sheet is comparatively weak. Make it stronger by indicating a direction (more likely/less likely for the existence).

l. 50 Which other ice grounding features? Where is this taken up again?

l. 86 A video in the Supplementary Info would be helpful to visualize this better.

l. 122 So would you say that the calving parameterization is more important than the buttressing effect of LR? I am missing a classification of all the different mechanisms throughout the text.

l. 129 This is a fairly significant change in sea level equivalent which is maybe worth mentioning in the abstract (near l. 14?).

l. 133 All model configurations lead to grounding on LR ridge (l. 133), yet there is no evidence that grounding occurred during the LGM (l. 96). This conflict needs to be resolved. Does it mean that your bracket for ocean melting is too low on the upper end?

l. 151 what was done with ocean melting, the East Siberian Ice Sheet and the other variable factors during these runs (this is emblematic shortcoming also in other paragraphs)?

l. 175 The 3.5 m mentioned earlier are a direct consequence of ice-shelf buttressing no?

l. 217 What sliding coefficient was assumed for LR? Does this matter?

l.225 Filschner --- Filchner

l. 258 I am missing an explanation if the CCSM4 climate forcing is impacted by the build-up of the large ice shelf.

Figure 1: Is quite similar to figure 1 in Nielsens et al., TC, 2017 (even the 1000 bathymetric contour is the same). Is that intentional to facilitate comparison? If so why not state that in the caption. The figure would benefit from a legend making it easier to assess which line is what. It should also be larger (especially the labels).

Figure 2d is referenced in the text, but a-c are not. Please check.

Figure 3: I must have missed it in the text, but why does calving go to 0 after 4 kyrs? What happens at the ice-shelf front after that? Also which melt enhancement factor was used for the lines displayed here (would it not be better to display the full range?).

Figure 4: Ice speed in meters per year? I think it should be kilometers per year. Please check. Also mark the location of LR here so that the reduction in flow velocities can be linked to it.

Figure S2: What does the legend stand for? Also what is shown is not the ice-shelf thickness (which should not be negative). Maybe better talk of an “ice-shelf profile”?

L. 482: “on off” → “on”

All Figures need a spatial scale.

Reference to Nielszen et al., 2017 should not reference the Discussion Paper but the peer-reviewed version.

Reviewer #3 (Remarks to the Author):

Review of Numerical simulations of a kilometre-thick Arctic ice shelf consistent with ice grounding observations by Gasson et al.

In 2014, bathymetry surveys of the Lomonosov Ridge proved the existence of the previously contentious idea of an Arctic-wide ice shelf during a past glaciation. The ice shelf had to have had considerable thickness in order to have grounded on the Ridge. This ice shelf did not exist during the last glaciation, but may have in the penultimate (MIS6) glaciation or earlier.

Gasson et al provide the first (as far as I am aware) 3D numerical ice sheet modelling study of an Arctic ice shelf. They are able to produce the ice shelf with a variety of ice sheet configurations. They show that a >1 km thick ice shelf can only form if the entire Arctic is covered in a shelf. They also show that the Lomonosov Ridge buttressing has an impact on the thickness and dynamics of the shelf (although not substantial). They also comment on the impact the Arctic ice shelf will have on global $\delta\text{-O18}$ values, which are commonly used as an indirect proxy of sea level and ice volume. The paper is in good shape, and I think this can be published with some minor revisions.

The authors use the highly mature ice sheet model by Pollard and DeConto, one of the current generation of models that combine the shallow shelf and shallow ice approximations. There are a lot of caveats to using a shallow approximation model for an ice shelf as large as the Arctic Ocean, but at present there is not really an alternative method that can cover the long time scales that are investigated in this study. The caveats of using a current generation ice sheet model are highlighted (and investigated in detail in the methods section). If I were to make a suggestion, it would be to explicitly state within the main text that there was no dynamical ocean impacts in this model. The present model uses subshelf melt rates that are based on simulations of Antarctic ice shelves, but I would assume that the circulation regime under an Arctic ice shelf would be substantially different, and that would have follow-on impacts on the evolution of the shelf. This parameter is very important, as shown in figure S4.

One of the conclusions made in this paper is that the LGM configuration Laurentide ice sheet was able to produce a thicker ice shelf than their smaller extent ice sheet (lines 151-169). I would like the authors to comment more on why this is in the main text, because that contradicts the geological evidence.

In addition, the authors use the ICE-5G 13 kyr ice sheet configuration for their "minimal" MIS6 configuration. Although the Keewatin sector of the Laurentide ice sheet was smaller in MIS6 (i.e. the ice sheet likely did not reach the Cordillera, and did not completely cover Banks Island), the eastern sector was larger than at the LGM (e.g. Curry et al 2011). This would likely have an impact on the climate model used in this paper, and may also impact the ice sheet model. I think a better justification for not making a MIS6 Laurentide margin limit reconstruction similar to the Eurasian reconstruction (especially considering that the speculative Siberian Ice Sheet is included) is needed, because I think that there is enough evidence to give a generalized MIS6 extent. I understand that the climate models are taken from another study, so I would ask is some text to explain what the likely impact of this might be. This might be one of the causes of the discrepancy mentioned in the previous paragraph. I do not think either of these factors will greatly impact the results presented in the paper, so I am not asking for more modelling runs.

Another question I have is why the authors chose not to show the modelled results of a reduced extent Laurentide Ice Sheet combined with a Siberian Ice Sheet. This may be one of the more important aspects of this kind of study, as the Siberian ice sheet did not exist during the LGM, but may have in MIS6. I assume this experiment was done (from the wording in the text), maybe it could be included as a supplement.

Minor Comments

Line 182: Can this value be given as a sea level equivalent as well?

Line 438: Reference 49 is missing some authors.

Fig 1: It is pretty hard to see the difference between the LGM and ICE-5G 13 kyr extents on this figure. Also, for figure 1b, it isn't stated in the caption which model setup was used.

General remark on figures: would it be possible to use more discrete intervals on the coloured plots? For instance, with the thickness plots, it is pretty hard to tell the difference in the range between 800 m and 1200 m thick ice, a distinction that is very important for this study. Maybe even some contours would be beneficial. The fonts on some figures could also be made a bit larger.

References

Curry, B.B., Grimley, D.A., McKay, E.D., 2011. Quaternary glaciations in Illinois, in: Ehlers, J., Gibbard, P.L., Hughes, P.D. (Eds.), Quaternary Glaciations - Extent and Chronology A Closer Look. Elsevier. volume 15 of Developments in Quaternary Sciences. chapter 36, pp. 467–487.

Regards,
- Evan J. Gowan

Reviewer #1 (Remarks to the Author):

Review of Gasson et al. NatureComm
“Numerical simulations of a kilometre-thick Arctic ice shelf consistent with ice grounding observations”

General

This is a really good new modeling study of the hypothesized Arctic Ocean-wide MIS thick ice shelf, and also perhaps a smaller ice shelf during the LGM. It is well written and the model results are plausible.

We thank the reviewer for their generally positive comments

But in places in the paper, the authors raise tantalizing topics (see specifics listed below) like rates of sea level rise, ocean circulation impacts of ice-shelf melting (ie page 8), calibration of foram O18 to ice volume, E Siberian Sea ice, etc, but they never really delve into them. These topics are huge in the eyes of NatComm readership. I would like to see more discussion of the implications of complete ice shelf cover during MIS 6, and post MIS 6 disintegration and likewise for other glacials, other than implications for the ice shelf modeling itself developed for Antarctica.

We agree with the reviewer that these are all interesting topics worthy of further study. We have added more discussion on sea level and d18O significance as well as new experiments addressing the potential for LGM ice shelves (a point also raised by Reviewers 2 and 3). However, we are not able to address all of the issues raised by the reviewer with the modeling experiments currently performed and presented in the manuscript, we discuss why below:

We are working towards coupled ice shelf – ocean simulations, but this is a major technical challenge. This is needed to address impacts of an ice shelf on ocean circulation in a more quantitative way. Previous publications have speculated on the role of thick ice in the Arctic on ocean circulation, in particular during the breakup of the ice shelf (e.g. Moore, 2005). We have reiterated this in the ‘Earth System Significance’ section, but do not want to overstate its significance until we have performed coupled ocean simulations.

We have added more text to the discussion on foram d18O to ice volume and sea level calibrations. We have estimated how the simulated ice shelf would affect the oxygen isotope composition of seawater. This is assuming a fixed d18O composition of the ice, which we have shown previously is a simplification (Gasson et al., 2016). We are working towards water isotope-enabled simulations of an Arctic ice shelf but these have not yet been performed and therefore are not included in this manuscript.

The authors also do not address the question of WHEN these massive ice sheet/shelf systems began: end Pliocene? Mid Pleistocene Transition, Mid-Brunhes Event? [a new paper on the MBE is coming out any day relevant to the MBE threshold in the Arctic].

We have added more text at the end of the introduction discussing when thick ice shelves may have formed:

It is possible that large Arctic ice shelves also formed during other glacial stages, although interestingly no erosional features deeper than ~600 m water depth have yet been dated to the LGM^{10,15}. Reconstructions of Arctic Ocean temperatures

suggest that, following the mid-Brunhes event (~400 ka), intermediate-depth temperatures were warmer than modern during glacial stages^{16,17}. One hypothesis is that a thickening halocline, driven by increased freshwater inputs and the formation of an Arctic ice shelf, caused the polar surface layer and inflowing warm Atlantic waters to deepen. Although there are considerable uncertainties, this would suggest that a thick ice shelf (but one that may not have grounded) formed during each of the last four glacial maxima¹⁷.

Finally are there any implications for modeling past and present Antarctic ice shelves and margins, since, if their Antarctic model is supported by all the empirical field evidence in the Arctic, this is an important conclusion.

Although this is an interesting idea, we feel that discussion of significance to Antarctic ice shelves may detract from the main message of the paper.

In Figures, Label ice sheets in Fig 1b, spell out Lomonosov Ridge, Amerasian, Eurasian Basin in Fig 1c. Fig 2 & fig. 4 have NO labeling, please add a little.

We have now added suggested labeling to Fig 1, 3 and 4. Although following Nature Communications figure guidelines we have tried to minimize text on figures.

In sum, perhaps the authors are understating the implications of their results for rapid deglaciation, SL rise, ice shelf collapse etc. Is the time framework shown in Figure 3 accurate, or, could there be more abrupt transitions that simply are not accounted for in the model structure. Recommendation: Publish with minor revision.

There may be more rapid transitions, which are not represented in the current simulations because we are using a constant climate forcing. This is now stated in the methods section.

Specifics

Line 34. Dating the break up at the end of MIS 6 is a hard thing to do [ie date], although Stein et al gave their best shot. This is critical and LGM ice break up is much better known due to the many C14 dated studies showing a widespread glacial unconformity and C14 dates mainly starting ~ 15 ka, the B/A most likely (Polyak et al. 2007, 2009, Poirier et al. 2012, Gemery et al 2017 Climate of the Past and many studies cited in these.

We have added a new section discussing MIS6 v. LGM ice shelves and the possibility that an ice shelf also formed during the LGM. We believe that discussing timing of LGM ice shelf retreat could be misleading to the reader, and overstate our confidence in the existence of an LGM Arctic ice shelf.

Line 38 are the ice streams in Figure 1 from Stokes' studies? Please cite here. – Yes they are, but we cite the original reference (Margold et al., 2015).

Line 42 spelling Arctic – Now corrected

Line 44. Though it may be covered later, it would be useful to say here up front if these models used include sub-ice ocean forcing or not. – Now stated here that there is not ice-ocean coupling

Line 47. More recent sea level and GIA evidence for E Siberian ice is found in Cronin et al. 2017 Climate of the Past (See also line 139) – Reference added

Line 69 whose N American and Eurasian grounding lines are you using here? – this is estimated from the model output.

Line 89-90 say ice shelf thickness or “its thickness” so as not to confuse with continental shelf - clarified

Line 96-99. Above there were citations to others favoring an LGM ice shelf cover, but here the idea is dismissed fairly quickly. See new section discussing potential for LGM shelf cover.

Line 103 you mean “which we reduced in the model simulation... ?? – now clarified

Line 154 we need more precise data on LGM & MIS 6 minimum sea level, from submarine regions. Agreed, there is added discussion on LGM and MIS6 sea level minima.

Line 172 the volume ... varies... ?? – corrected

Line 178 awkward sentence – agreed, now changed

Line 180-181, is this corrected for deep bottom temperature, a large source of MIS 6 oxygen isotope variability/signal. – no, this is the total shift observed in benthic foraminifera, we have now clarified that this is both a deep sea temperature and ice volume signal.

Line 198- PLEASE Say how rapid and do estimates jive with rates of SL rise? – We have not included deglacial simulations in the manuscript so do not want to speculate on rates of retreat. In preliminary simulations that we have performed the breakup is rapid (rates similar to MWP-1a), however this is driven in part by strong basal melting in addition to calving. As we are not using a coupled ice-ocean model at present we do not know whether this strong basal melting is a result of the imposed ocean temperatures or not. We agree that this is an extremely interesting area of future research. The rapid calving we mention in the final sentence.

Reviewer #2 (Remarks to the Author):

Gasson et al. investigate the possibility of a thick Arctic ice shelf covering the entire Arctic basin during the penultimate glacial maximum (and maybe also during the LGM). They do this by using an SSA-SIA model with a number of different boundary conditions/forcings. Some of these stem from climate models (e.g. the surface mass balance) and paleo-reconstructions (e.g. the ice-sheet extents), others are treated in a more ad-hoc way (e.g. ocean melting and calving thresholds). Within this [huge] parameter space, they find stable configurations for such a large ice shelf. A key constraint for evaluating the different model runs are recently discovered geomorphological features at the Lamasov ridge (and on other ridges equally far away from the continental grounding line) which are supposedly caused by erosion from a locally re-grounded ice shelf. Smaller ice-shelves (i.e. restricted in size to the Amerasian basin) would not develop the required thickness for re-grounding, which is used as an argument to rule out a previously hypothesized ‘minimum-model’.

I have read this paper with great interest, and without doubt the question whether or not such a large ice shelf has existed has important consequences for a number of

disciplines such as glaciology, geomorphology, oceanography and sea-level reconstructions. The timely manner of this problem is demonstrated by a number of publications (e.g. Stein et al. 2017, Jakobsen et al. 2016, Nielsen et al. 2017) which have all been published recently. The merit of this study over previous modelling approaches (e.g. Nielsen et al. 2017, TC) is that it uses a real geometry and observationally constrained forcings which surely is a complicated task and I compliment the authors for achieving this.

I encourage the authors to put more effort into highlighting the critical mechanisms and to order them in importance (if possible). At the moment, the paper partially invokes the impression that some scenarios have been tailored (e.g. by reducing the calving threshold) so that such a large ice shelf can exist and explain the observed glacial landforms. While this in itself is also an important result, it undersells the potential of the study for other researchers.

The different model scenarios must be explained more clearly (i.e. which parameters have been varied simultaneously or individually and over what range). Currently I fear that the study will not be reproducible with the information supplied in the text. Nature Communications has no tight space limits so more plots/text can easily be added.

We thank the reviewer for their constructive comments. To aid reproducibility, tables of all experiments have been added. These also include the sensitivity tests. We have tried to make it clearer in the text that the equilibrium ice shelf thickness is strongly influenced by the ocean melt rate, in particular.

Below, I have a number of comments/questions (some of which may reflect my inexperience with paleo ice sheet modelling). These comments should be considered and I hope that you will find them helpful in improving the current version of the paper.

Kind regards, □ Reinhard Drews, University of Tübingen, Germany

Again, we thank the reviewer for their comments and believe that they have improved the manuscript.

General Comments □ 1. The reduction of the calving threshold seems a necessity to grow this large ice shelf. It is motivated in a few lines (l. 60-62) with the BKIS which does – in this model -- not grow to its known LGM extent. However, does the reduction of the calving threshold have any physical motivation (other than maybe accounting for sea-ice cover/ice mélange). Why would calving be different in the Arctic than in the Antarctic? How can you tell that the too small BKIS extent is not due to other factors such as erroneous forcing (e.g. a too small surface mass balance or too high ocean melting?). I suggest to argue more strongly that the reduction of the calving threshold is physically reasonable and is not used as a tuning parameter correcting for a bias in the input data (or for the simplified model physics of the SIA-SSA model). In the main manuscript and methods we have now clearly stated that formation of the ice shelf is dependent on a reduction in h_{calv} . The justification is that it is required to simulate regrowth of the WAIS and formation of the BKIS. For Antarctica, either h_{calv} is removed or calving is reduced in confined embayments, attributed to clogging with ice mélange and/or sea ice. We have removed this parameterization as it would treat the entire Arctic Basin as a confined basin and instead lower h_{calv} . We have now explained this more clearly in the methods. Simulation of ice mélange and its impact on calving is the subject of future study.

We have tested whether the lack of BKIS is a result of the ocean melt rate and have found that it is not. In simulations with the higher calving threshold (150 m) and low ocean melt the BKIS is also prevented from forming.

2. A sensitivity study has been performed, but it appears scattered throughout the text and important information is missing (or I missed it). Parameters include bathymetry with and without LR, melt rates with enhancement factor ranging from 0.5-8, climate input from LGM and PGM, two different extents of the Laurentide Ice Sheet, and existence/absence of an East Siberian Ice Sheet. However, it is unclear to me which parameters were varied in which run. I suggest to summarize the sensitivity studies in a table, and to name the different model runs correspondingly, otherwise it is very hard to get an overview of what was done. How many model runs were done in total? A good suggestion, we have added a table summarizing all experiments in the main text.

3. The previous points highlights are more important issue: Which of all these parameters is most critical? Which one is the most ill-constrained? (Is it the missing calving-law, the ice-ocean coupling, the bathymetry?). This paper currently gives little outlook/motivation for other researchers to redirect their research focus. We hope that the inclusion of the new table will make it more clearer what impact the different parameters have on the results.

4. The authors loosely use the term “pinning” for the grounding of the ice shelf on LR and I think that they mean by that the development of ice rumples (as opposed to ice rises). However, what basal friction coefficient has been assumed for the LR and does this value matter? As a function of this, what are the basal sliding velocities? Pinning points in Antarctica can be near stagnant (e.g. Berger et al., 2016, The Cryosphere) or almost fully overridden (e.g. the Doake Ice Rumples). At which end of the spectrum do you see the LR? Without those details the statement that LR has “limited impact on ice- shelf dynamics (l. 14)” is hard to assess. We would consider the feature on the central LR an ice rise rather than a rumples, with low basal velocities and elevations >100 m above the surrounding ice shelf. We have clarified that we are referring to large-scale shelf dynamics.

5. All but the (ruled-out) minimum-model configurations lead to grounding on LR ridge (l. 133), yet there is no evidence that grounding occurred during the LGM (l. 96). This conflict needs to be resolved. Does it mean that your bracket for ocean melting is too low on the upper end?

Thank you for this point. We have now added an additional simulation for the LGM and addressed the issue of MIS6 v LGM ice shelves. There is no evidence for grounding below 600 m for the LGM; this is something that cannot be explained with our existing experiments, in which we test how the different configurations of terrestrial ice sheets could affect the ability for an Arctic ice shelf to form. Within our ocean melting sensitivity tests, an enhancement factor of 4 can produce an ice shelf that does not ground on the Lomonosov Ridge. We therefore suggest that differences in ocean circulation could explain why a thick ice shelf did not form during the LGM.

6. How important is the surface mass balance to sustain the ice shelf (e.g. compared to the basal melt rates)? How reliable are those values, and how does the existence of the large Arctic ice shelf impact the surface mass balance rates (is there some kind of Feedback mechanism?)

The relative roles of ocean melting and surface mass balance are shown in Figure 3 and Table 2. Although the GCM simulation does not include an ice shelf, it does have a sea ice cover in the Arctic, so we do not expect this difference to have a large impact on the surface mass balance via feedback mechanisms. We have stated more clearly that the GCM simulations do not include an ice shelf.

7. Do the ocean melt rates applied here bracket the estimations done by Jakobsson et al. (NComms, 2016)? Melt rates of max 1.27 m a^{-1} at 1000 m depth seem not overly high (although an ocean enhancement factor of 8 sounds high). It is important to put this into perspective with other studies because ocean melting is clearly critical to ice-shelf stability. □

The values estimated in the analysis of Jakobsson are comparable. Text added:

For the default melt enhancement factor, used in all experiments in the main manuscript, basal melt rates at 1000 m depth are $\sim 0.16 \text{ m yr}^{-1}$ in the central Arctic. This is comparable to the basal melt rates of $< 0.2 \text{ m yr}^{-1}$ estimated by Ref 10 using a conceptual oceanographic model

8. I suggest to show all the input parameters of a minimum (smallest ice sheets, highest melting, highest calving threshold, no East Siberian Ice Sheet) versus a maximum (largest ice sheets, smallest basal melting, smallest calving threshold, East Siberian Ice Sheet) scenario in a supplementary Figure. This would also make Fig. 1 more accessible (which contains some of this information but not all).

All input parameters that are varied are now shown in a table, which also shows impact on shelf thickness.

I. 16: "Significance" for the hypothesize East Siberian Ice Sheet is comparatively weak. Make it stronger by indicating a direction (more likely/less likely for the existence). *We have changed this to: **An Arctic Ice Shelf could have formed even in the absence of a hypothesized East Siberian Ice Sheet***

I. 50 Which other ice grounding features? Where is this taken up again? *We discuss grounding on the Arlis Plateau later in the text. A table has been added summarizing all locations of ice grounding.*

I. 86 A video in the Supplementary Info would be helpful to visualize this better.

We have now added a video showing shelf formation.

I. 122 So would you say that the calving parameterization is more important than the buttressing effect of LR? I am missing a classification of all the different mechanisms throughout the text. *Clarified that the LR is less important, as the shelf can form in a simulation in which the bathymetry is lowered to remove the LR.*

I. 129 This is a fairly significant change in sea level equivalent which is maybe worth mentioning in the abstract (near I. 14?). *Agreed, this is quite a large change, although we would prefer to avoid adding too many figures to the abstract.*

I. 133 All model configurations lead to grounding on LR ridge (I. 133), yet there is no evidence that grounding occurred during the LGM (I. 96). This conflict needs to be resolved. Does it mean that your bracket for ocean melting is too low on the upper

end? We have added new discussion on LGM v. MIS6 ice shelves, this conflict has not been resolved and we suggest this as an area for future study.

I. 151 what was done with ocean melting, the East Siberian Ice Sheet and the other variable factors during these runs (this is emblematic shortcoming also in other paragraphs)? As suggested, a table of all experiments is now included.

I. 175 The 3.5 m mentioned earlier are a direct consequence of ice-shelf buttressing no? Clarified that it will impact sea level through buttressing.

I. 217 What sliding coefficient was assumed for LR? Does this matter? Deformable sediment. We do not believe this will have a significant impact on the results.

I.225 Filschner --- Filchner Thanks for spotting

I. 258 I am missing an explanation if the CCSM4 climate forcing is impacted by the build-up of the large ice shelf. Now state clearly: *These simulations do not include an Arctic ice shelf and there is therefore no feedback from the formation of an ice shelf on the ocean and climate system.*

Figure 1: Is quite similar to figure 1 in Nielszen et al., TC, 2017 (even the 1000 bathymetric contour is the same). Is that intentional to facilitate comparison? If so why not state that in the caption. The figure would benefit from a legend making it easier to assess which line is what. It should also be larger (especially the labels).

Although we agree that they are similar, this is not intentional and the figure was made prior to publication of Nilsson 2017. The directional arrows are from Jakobsson 2016, which is cited in the figure caption. We are happy to make this larger as a single figure, but would prefer to keep the parts A and B together. We have added a legend showing what the orange and black lines represent.

Figure 2d is referenced in the text, but a-c are not. Please check. We now reference a-c in the text.

Figure 3: I must have missed it in the text, but why does calving go to 0 after 4 kyrs? What happens at the ice-shelf front after that? Also which melt enhancement factor was used for the lines displayed here (would it not be better to display the full range?). Clarified that this is for the Arctic Basin region only and that calving is zero when the calving front has moved outside of the basin. The table of simulations is now added to show what melt enhancement factor is used.

Figure 4: Ice speed in meters per year? I think it should be kilometers per year. Please check. Also mark the location of LR here so that the reduction in flow velocities can be linked to it. Apologies, this is a log10 scale, now corrected.

Figure S2: What does the legend stand for? Also what is shown is not the ice-shelf thickness (which should not be negative). Maybe better talk of an "ice-shelf profile"? Now corrected

L. 482: "on off"à"on"□ - unsure what this refers to. All Figures need a spatial scale. We have added latitude and longitude markings where appropriate

Reference to Nielszen et al., 2017 should not reference the Discussion Paper but the peer- reviewed version. Thank you, now corrected.

Reviewer #3 (Remarks to the Author):

Review of Numerical simulations of a kilometre-thick Arctic ice shelf consistent with ice grounding observations by Gasson et al.

In 2014, bathymetry surveys of the Lomonosov Ridge proved the existence of the previously contentious idea of an Arctic-wide ice shelf during a past glaciation. The ice shelf had to have had considerable thickness in order to have grounded on the Ridge. This ice shelf did not exist during the last glaciation, but may have in the penultimate (MIS6) glaciation or earlier.

Gasson et al provide the first (as far as I am aware) 3D numerical ice sheet modelling study of an Arctic ice shelf. – Note that Colleoni et al., 2016 (Quaternary Science Reviews) simulated an East Siberian ice shelf – this is cited in the text.

They are able to produce the ice shelf with a variety of ice sheet configurations. They show that a >1 km thick ice shelf can only form if the entire Arctic is covered in a shelf. They also show that the Lomonosov Ridge buttressing has an impact on the thickness and dynamics of the shelf (although not substantial). They also comment on the impact the Arctic ice shelf will have on global del-O18 values, which are commonly used as an indirect proxy of sea level and ice volume. The paper is in good shape, and I think this can be published with some minor revisions.

We thank the reviewer for their comments and constructive suggestions

The authors use the highly mature ice sheet model by Pollard and DeConto, one of the current generation of models that combine the shallow shelf and shallow ice approximations. There are a lot of caveats to using a shallow approximation model for an ice shelf as large as the Arctic Ocean, but at present there is not really an alternative method that can cover the long time scales that are investigated in this study. The caveats of using a current generation ice sheet model are highlighted (and investigated in detail in the methods section). If I were to make a suggestion, it would be to explicitly state within the main text that there was no dynamical ocean impacts in this model. The present model uses subshelf melt rates that are based on simulations of Antarctic ice shelves, but I would assume that the circulation regime under an Arctic ice shelf would be substantially different, and that would have follow-on impacts on the evolution of the shelf. This parameter is very important, as shown in figure S4.

- We now state in the main text that the model does not have dynamical ice-ocean coupling. Agreed, this is an important area for future study.

One of the conclusions made in this paper is that the LGM configuration Laurentide ice sheet was able to produce a thicker ice shelf than their smaller extent ice sheet (lines 151-169). I would like the authors to comment more on why this is in the main text, because that contradicts the geological evidence.

- We have now addressed the issue of LGM v MIS6 Arctic ice shelves and have performed an additional experiment with an LGM configuration. As suggested by the reviewer a thick ice shelf also forms during the LGM in our simulations, contradicting the current geological evidence. We suggest that differences in ocean circulation, and not the configuration of the terrestrial ice sheets, may explain why a thick ice shelf did not form during the LGM.

In addition, the authors use the ICE-5G 13 kyr ice sheet configuration for their “minimal” MIS6 configuration. Although the Keewatin sector of the Laurentide ice sheet was smaller in MIS6 (i.e. the ice sheet likely did not reach the Cordillera, and did not completely cover Banks Island), the eastern sector was larger than at the LGM (e.g. Curry et al 2011). This would likely have an impact on the climate model used in this paper, and may also impact the ice sheet model. I think a better justification for not making a MIS6 Laurentide margin limit reconstruction similar to the Eurasian reconstruction (especially considering that the speculative Siberian Ice Sheet is included) is needed, because I think that there is enough evidence to give a generalized MIS6 extent. I understand that the climate models are taken from another study, so I would ask is some text to explain what the likely impact of this might be. This might be one of the causes of the discrepancy mentioned in the previous paragraph. I do not think either of these factors will greatly impact the results presented in the paper, so I am not asking for more modelling runs.

We have added more text discussing the extent of the Laurentide during the LGM and MIS6 (see below). We do not believe there is sufficient evidence (and do not know of an available reconstruction) to constrain the MIS6 Laurentide ice sheet in the same way that we have done for the Eurasian ice sheet. Developing such an outline is well beyond the scope of this study. We have explored whether constraining the Laurentide ice sheet to its LGM extent (using the Dyke et al., 2002 margin) impacts the results. This does slightly reduce ice flow into the basin, largely because in our simulations we have ice cover around Banks Island, and reduces the ice shelf thickness slightly. This does not impact the overall conclusions of the paper, and we would prefer not to repeat all of the simulations with this constraint included.

Two different extents of the Laurentide Ice Sheet are considered in the climate model forcing (the ice sheet model margin is not constrained), one is equivalent to the LGM extent, and one with a reduced extent equivalent to the ICE5G reconstruction at 13 ka (following Ref 22). For the LGM the extent of the major ice sheets is well known, constrained by a large volume of dated geomorphological and geological evidence^{27,28}. For prior glaciations such as MIS6, the evidence-base in North America is sparse and its geographic spread is insufficient to develop a continent-wide reconstruction of ice extent at this time. But it is known for example that during MIS6, that the southern Laurentide Ice Sheet extended some 200 km further south in the US state of Illinois than during the LGM²⁹. In Ohio the MIS6 extent is a short distance (10-20 km) further south³⁰. In Minnesota, pre-LGM glacial deposits also extend beyond the known LGM ice limits³¹. However, this picture of greater MIS6 extent is not repeated around the rest of the perimeter of the ice sheet, likely because the ice sheet was smaller than during the LGM and the evidence has been erased or obscured. Sea level estimates suggest that the LGM sea level lowstand was lower than during MIS6, with one estimate suggesting a difference between the two glacial stages of 21 ± 14 m³³. Additionally, because the MIS6 Eurasian ice sheet had a greater extent than during the LGM³⁴, it would also suggest that the North American Ice Sheets had a lower volume during MIS6^{23,35}.

Another question I have is why the authors chose not to show the modelled results of a reduced extent Laurentide Ice Sheet combined with a Siberian Ice Sheet. This may be one of the more important aspects of this kind of study, as the Siberian ice sheet did not exist during the LGM, but may have in MIS6. I assume this experiment was done (from the wording in the text), maybe it could be included as a supplement.

This experiment has now been performed at added to Table 1

Minor Comments

Line 182: Can this value be given as a sea level equivalent as well? Now added

Line 438: Reference 49 is missing some authors. Corrected

Fig 1: It is pretty hard to see the difference between the LGM and ICE-5G 13 kyr extents on this figure. Also, for figure 1b, it isn't stated in the caption which model setup was used. Agreed, have removed these from the figure and replaced with Dyke 2003 outline. The two GCM configurations are now included in a supplementary figure.

General remark on figures: would it be possible to use more discrete intervals on the coloured plots? For instance, with the thickness plots, it is pretty hard to tell the difference in the range between 800 m and 1200 m thick ice, a distinction that is very important for this study. Maybe even some contours would be beneficial. The fonts on some figures could also be made a bit larger.

We have attempted to make the figures clearer by reducing the number of colour intervals. We decided against adding contours as the ice shelf thickness is fairly uniform, so this does not improve clarity. The mean shelf thickness is included in the figure caption and table.

References

Curry, B.B., Grimley, D.A., McKay, E.D., 2011. Quaternary glaciations in Illinois, in: Ehlers, J., Gibbard, P.L., Hughes, P.D. (Eds.), Quaternary Glaciations - Extent and Chronology A Closer Look. Elsevier. volume 15 of Developments in Quaternary Sciences. chapter 36, pp. 467–487.

Regards,
- Evan J. Gowan

REVIEWERS' COMMENTS:

Reviewer #1 (Remarks to the Author):

The authors adequately revised the manuscript to address my own and other reviewers' questions. The paper is very clearly written and will be influential. I recommend publishing as is.

Reviewer #2 (Remarks to the Author):

From my point of view, the authors have provided a comprehensive response to all the concerns raised during the first round of reviews. Limitations of the study are now clearly mentioned (e.g. the missing ice-ocean coupling), and the authors are careful not to oversell their implications, for example, for sea-level rise reconstructions and ocean circulation patterns. The existence (or absence) of such a large ice shelf in the Arctic is clearly interesting for a wide readership and a number scientific disciplines.

I have no substantial objections for publishing this study, and I compliment the authors for a substantial and meaningful piece of work.

Kind regards,
Reinhard Drews, University of Tübingen, Germany

Minor comments

- I. 30 Consider adding age range in ka (as done below for the mid-Brunhes event)
 - I. 87 should it be „an Eurasian Ice Sheet“ (instead of „a Eurasian Ice Sheet)
 - I. 105 unify the use of "ka" and "kyrs"
 - I. 126 „East Siberia“ (instead of „East Siberian“)
 - I. 215 The simulations without the LR are helpful and clearly show that some of my earlier remarks regarding the basal friction coefficient at LR are not critical for the formation of the large ice shelf.
 - I. 277 I am no expert in these grounding-line parameterization schemes, but now, or at some later stage, this grounding line scheme (Schoofing?) should be discussed in the light of the findings from Reese et al. (TCD, 2017) assuming successful peer-review.
 - I. 370 maybe „poorly constrained today, and even worse in the past.“
- Table 1 – Label the first line (numbers 1 -9).

Reviewer #3 (Remarks to the Author):

I have read through the responses and the revised manuscript, and I am satisfied with the authors efforts to address the comments by all three reviewers. I think the figures are definitely more clear than in the first version, and I like the the addition of tables describing the outcomes of the experiments performed. I have a couple of minor comments:

- 1) References 34 and 40 (Rohling et al) are the same.
- 2) The font used in Figure 2 looks very squished. It might be a good idea to modify it to have a consistent presentation as the other figures.

Best Regards,
Evan J. Gowan

REVIEWERS' COMMENTS:

Reviewer #1 (Remarks to the Author):

The authors adequately revised the manuscript to address my own and other reviewers' questions. The paper is very clearly written and will be influential. I recommend publishing as is.

Thank you for these comments. We thank the reviewer for their time and comments that have greatly improved the manuscript.

Reviewer #2 (Remarks to the Author):

From my point of view, the authors have provided a comprehensive response to all the concerns raised during the first round of reviews. Limitations of the study are now clearly mentioned (e.g. the missing ice-ocean coupling), and the authors are careful not to oversell their implications, for example, for sea-level rise reconstructions and ocean circulation patterns. The existence (or absence) of such a large ice shelf in the Arctic is clearly interesting for a wide readership and a number scientific disciplines.

I have no substantial objections for publishing this study, and I compliment the authors for a substantial and meaningful piece of work.

We really appreciate these comments. Again, we thank the reviewer for the time taken to review this manuscript and the various suggestions that have improved the manuscript.

Kind regards,
Reinhard Drews, University of Tübingen, Germany

Minor comments

l. 30 Consider adding age range in ka (as done below for the mid-Brunhes event) – Now added

l. 87 should it be „an Eurasian Ice Sheet“ (instead of „a Eurasian Ice Sheet)
We believe 'a Eurasian Ice Sheet' is correct.

l. 105 unify the use of "ka" and "kyrs" – unified throughout, ka for a age before present, kyr for a time interval.

l. 126 „East Siberia“ (instead of „East Siberian“) - corrected

l. 215 The simulations without the LR are helpful and clearly show that some of my earlier remarks regarding the basal friction coefficient at LR are not critical for the formation of the large ice shelf. agreed

l. 277 I am no expert in these grounding-line parameterization schemes, but now, or at some later stage, this grounding line scheme (Schoofing?) should be discussed in the light of the findings from Reese et al. (TCD, 2017) assuming successful peer-review. – This is an interesting point. We agree that the Reese paper is certainly of interest but have decided not to address this in

the current article until that paper has completed review.

l. 370 maybe „poorly constrained today, and even worse in the past.“ - added
Table 1 – Label the first line (numbers 1 -9). – now labeled.

Reviewer #3 (Remarks to the Author):

I have read through the responses and the revised manuscript, and I am satisfied with the authors efforts to address the comments by all three reviewers. I think the figures are definitely more clear than in the first version, and I like the the addition of tables describing the outcomes of the experiments performed. I have a couple of minor comments:

We thank the reviewer for their time and comments on the manuscript.

1) References 34 and 40 (Rohling et al) are the same. Thanks for spotting, now corrected

2) The font used in Figure 2 looks very squished. It might be a good idea to modify it to have a consistent presentation as the other figures. We have changed the font size on Figure 2.

Best Regards,
Evan J. Gowan